# TT Genotype of *TLR4 rs1928295* Is a Risk Factor of Overweight/Obesity in Han Chinese Children Aged 7–12 Years and Can Interact with Dietary Patterns to Affect the Incidence of Central Obesity and Lipid Profile, Systolic Blood Pressure Levels

**DOI:** 10.3390/nu15153441

**Published:** 2023-08-03

**Authors:** Qi Zhu, Ben Qian, Kun Xue, Hongwei Guo, Rui Liang, Jinlong Wu, Qisu Wu, Geyi Zhou

**Affiliations:** 1School of Public Health, Nantong University, Nantong 226019, China; zgjsntzq@126.com (Q.Z.); 15996538250@163.com (Q.W.); zgyyfyxsspa@163.com (G.Z.); 2Department of Medicine, Xinglin College, Nantong University, Nantong 226019, China; 3Department of Nutrition, School of Public Health, Fudan University, Shanghai 200032, China; xuekun@shmu.edu.cn

**Keywords:** *TLR4 rs1928295* polymorphisms, children obesity, dietary patterns, macronutrients, interaction analysis

## Abstract

Previous studies have found that *TLR4 rs1928295* polymorphism is associated with Body Mass Index in European and American Indian adults. This study evaluates the relationship between this locus polymorphism, obesity-related parameters and dietary patterns in Chinese Han Children. A total of 798 children aged 7–12 years were included in this cross-sectional study. An improved Multiple Ligase Detection Reaction was used for genotyping. Dietary patterns were identified by principal component factor analysis. The overweight/obesity rate of the TT genotype was greater than those of the CC/CT genotype (*p* = 0.032 and 0.048 in boys and girls, respectively). Boys of the TT genotype could interact with protein and cholesterol intake to increase low density lipoprotein (LDL) levels (*p* = 0.02, 0.015, respectively), while girls of the TT genotype could interact with total energy intake to increase triglyceride (TG) (*p* = 0.018) levels. Boys predisposed to a healthy balance dietary pattern (HBDP) and girls predisposed to an egg/fruit/fish dietary pattern (EFDP) were significantly associated with lower rates of central obesity (*p* = 0.045, 0.028). Boys carrying the TT genotype and predisposed to animal food dietary pattern (AFDP) had a higher level of low-density lipoprotein (*p* = 0.017) and systolic pressure (*p* = 0.044). Our results indicated that the TT genotype of *TLR4 rs1928295* is a potential risk factor for obesity in Chinese Han children and is associated with dietary patterns.

## 1. Introduction

The prevalence of overweight and obesity has been an issue of public health, endangering the well-being of children and adolescents worldwide. According to a recent study, the combined prevalence rate of overweight/obesity among school-age children and adolescents (7–17 years old) in China will reach 31.8% in 2030 [1]. Childhood obesity can lead to both physical and psychological diseases, and bring a variety of harm to the cardiovascular system, endocrine system, respiratory system, bone, liver, psychological, and cognitive intelligence [2], so it is very important to further study childhood obesity and formulate relevant strategies.

Obesity is a complex, multi-factorial, and heterogeneous disease [3], involving the interaction between environmental and genetic risk factors [4]. The most common type of human genetic variation are single nucleotide polymorphisms (SNPs), which account for more than 90% of all known polymorphisms. Widely studied polymorphisms of *FTO*, *MC4R,* and *LEPR* genes have been found to be correlated with higher total lipid and energy consumption, and interact with protein intake, resulting in weight loss or gain [5].

With the further development of gene-diet research, more and more gene polymorphisms have been found to interact with diet patterns. For example, the non-g-allele of *ADIPOQ* (adiponectin) *rs3774261* could interact with the Mediterranean low-calorie diet pattern to significantly improve lipid profile [6]. The T allele of *RETN* (resistin) *rs10401670* could interact with a low calorie, high fat diet and is significantly connected with insulin resistance and triglycerides [7]. 

The same SNP may differ between races. A study on *TAS2R38* bitter taste gene polymorphism found that AVI/AVI diploid was associated with a higher risk of obesity in European Americans and Asian Americans but had no significant effect in African Americans [8]. Therefore, the effect of SNPs on obesity should be studied separately for different ethnic groups. Of China’s 56 ethnic groups 90% are Han, thus it is important to study the correlation and interaction between dietary nutrients and single nucleotide polymorphisms in order to prevent and intervene in obesity in Han children of China.

Toll-like receptor (TLR) proteins are a class of innate immune receptors expressed on the cell membrane and related to microbial recognition, including 13 subtypes. TLR4 is the most important signal receptor mediating the immune response to bacterial lipopolysaccharide. In addition, TLR4 signaling might also be activated by SFAs and increase inflammatory cytokine production [9]. Under the conditions of a high fat diet, functional TLR4 is one of the necessary conditions to induce obesity [10]. Obesity is a kind of chronic low-grade inflammation [11]; hypothalamic inflammation is an important step in the initiation of insulin resistance caused by obesity, and inflammation is associated with many epigenetic changes, thus causing alterations in immune cell function that play an important role in the formation of obesity [12], with special emphasis on the role of the TLR4 signaling pathway [13]. Among them, TLR4 rs4986790 gene polymorphism has been linked with systolic blood pressure level [14,15], insulin resistance, and protein expression in obese patients [16]. 

According to a 2015 meta-analysis of GWAS and Metabochip based on a total of 339,224 subjects (322,154 European and 17,072 non-European), a significant correlation was found between body mass index (BMI) and the gene locus *TLR4 rs1928295* [17]. No study has been conducted on the connection between overweight/obesity and *TLR4 rs1928295* polymorphisms. The research on *TLR4* gene polymorphism in the Chinese population is also very scarce, and there are only relevant studies on Henoch–Schonlein purpura [18], sporadic Parkinson’s disease [19], and gout [20]. Similar to the present study, another locus, *TLR4 rs2149356,* was not found to be associated with obesity in the Chinese population, and the analysis was not combined with dietary factors [21]. So, we chose to explore the novel locus *TLR4 rs1928295*, which would be an important supplement to the research on TLR4 gene polymorphism of overweight/obesity occurrence in China. This study aims to preliminarily reveal the link between *TLR4 rs1928295* gene polymorphism and Chinese Han school-age (7–12 years) children’s overweight/obesity, and further explore the interaction between *TLR4 rs1928295* polymorphism, macronutrient intake, and dietary patterns on the risk of overweight/obesity.

## 2. Materials and Methods

### 2.1. Research Population 

This research was attached to NISCOC (nutrition-based comprehensive intervention study on childhood obesity in China): a randomized, cluster-controlled study conducted at multiple sites (Chinese Clinical Trials Registry (the primary registry of the WHO Registry Network) identifier: ChiCTR-TRC-00000402). A baseline survey was conducted at the Shanghai Center to collect these data. A stratified random cluster sampling method was used to ensure the representativeness of the samples. First, the primary school was separated into three layers: rural, suburban, and urban strata. Two schools were randomly selected from each layer. After that, two classes (grades 1–4, years 7–12) from each grade were selected at random. Between 30 and 40 students from the class were selected as study participants. In total, the number of healthy Chinese Han children who participated in the survey was 1511. The Ethics Review Committee of the Institute of Nutrition and Food Safety of the Chinese Center for Disease Control and Prevention approved this study (ethics number: 20081201). All students, parents, or legal guardians participating in this project have signed an informed consent form after being informed of the purpose and procedures of this study.

### 2.2. Anthropometric Assessments

We used an electronic column scale (GMCS-I; Yishen, Shanghai, China) to measure fasting weight and height. Meanwhile, participants wore minimal clothing and participated barefoot. More than 0.1% accuracy was achieved by the instrument. BMI was measured by weight (kg)/height squared (m^2^). Overweight/obese subjects were screened by BMI according to the ‘Screening for overweight and obesity in school-age children and adolescents’ released by the National Health and Family Planning Commission of the People’s Republic of China in 2018 [22]. Waist circumference (WC) was measured when normal exhalation ended with an inelastic tape between the lowest costal margin and the iliac crest, with no pressure on the body surface, with a precision of 0.1 cm. Participants who were obese should be measured near the most obese location. Measurements were repeated twice and then averaged. Divide waist circumference by height to calculate waist-to-height ratio (WHtR); the participants whose WHtR were greater than 0.45 in female children and 0.47 in male children would be defined as central obesity [23]. Blood pressure was measured using a mercury sphygmomanometer after the participants were placed in a quiet room and rested for at least 10 min before the measurement. The child’s sleeve was rolled up, exposing the arm on the table, so that the upper arm was level with the heart. The appropriate width of the cuff, about 1/2 to 1/3 of the upper arm circumference, was selected according to the child’s upper arm circumference. The cuff of the blood pressure monitor should be wrapped around the side of the child’s upper arm. The lower edge of the cuff should be above the elbow joint, and the tightness should be appropriate to ensure careful auscultation and accurate reading.

### 2.3. Diet Assessment

We used a standardized sheet of food record with food size labels (including vegetables, fruits, fish, eggs, milk, rice, wheat, potatoes, nuts, pork, poultry, pastries, confectionery, beans and their products, the 14 categories. The type and amount of food consumed by participants was recorded with the assistance of parents and guardians. Duration was three consecutive days (including two days and a weekend). The investigator explained and demonstrated the method of recording daily meal types and intakes to the respondents, investigated the dietary intake on the same day, and distributed questionnaires to the respondents to record their dietary intake in the following two days. 

A self-developed meal recipe analysis software (using the authoritative data from the Chinese Food Composition edited by the Institute of Nutrition and Food Safety, the Chinese Center for Disease Control and Prevention) was used to convert the dietary survey results into energy and nutrient intake. All foods were classified into 14 food types for subsequent extraction of dietary patterns, that is, the type, proportion, or combination of different foods in the daily diet. 

### 2.4. Laboratory Assays

With the cooperation of medical staff, 10 mL of fasting venous blood was collected, and serum was separated. The levels of triglycerides (TGs), total cholesterol (TC), high-density lipoprotein (HDL), low-density lipoprotein (LDL), and glucose (FBG) in the serum were measured. All samples were measured by standard methods (Enzyme kit: Fenghui Medical Technology Company, Shanghai, China; LX-20 automatic biochemical analyzer: Beckman Clouter, CA, USA).

### 2.5. Genotyping

Use the phenol chloroform method to extract DNA from whole blood. The *TLR4 rs1928295* SNP was genotyped by using the improved Multiple Ligase Detection Reaction (iMLDR). *TLR4rs1928295* (GENESKY Biotechnology, Shanghai, China) PCR primers were used as follows: forward 5′-GAAGTTCCAAGTACTGCCAGGGATAG-3′; reverse 5′-TTGAGAGCTGCCCACACACT-3′ (Sangon Biotech, Shanghai, China). The final volume of the PCR reaction was 20 µL, containing 1 µL extracted DNA, 1 µL primer, 1* GC-I buffer (BGRI Biological Technology, Shanghai, China), 3.0 mM Mg^2+^, 0.3 mM dNTP (Generay Biotech, Shanghai, China), and 1 U HotStarTaq polymerase (Qiagen, Dusseldorf, NRW, Germany), which was passed through a DNA thermal circulator with the following conditions: the DNA template was denatured at 95 °C for 2 min; amplification at 94 °C for 20 s; 65 °C for 40 s (−0.5 °C/cycle); 72 °C for 1.5 min was performed for a total of 11 cycles, 24 cycles at 94 °C for 20 s, 59 °C for 30 s, 72 °C for 1.5 min; and 72 °C for a final extension of 2 min. DNA was digested and amplified (20 µL) using 5 U SAP (Promega, Madison, WI, USA) and 2 U Epicentre (US) enzymes at 37 °C for an hour which were then inactivated at 75 °C for 15 min. Ligase chain reaction (LCR) was performed using the following primers: RC:

TTCCGCGTTCGGACTGATATGCTGCCCACACACACTAAGGCTGAG; RT: TACGGTTATTCGGGCTCCTGTGCTGCCCACACACACTAAGGCTGAA; RP:

GACACTGGGAAAGGCAGACTTACC (Sangon Biotech, Shanghai, China). The LCR reaction system consisted of 1 µL 10* bonding buffer, 0.25 µL high temperature ligase (Thermo Fischel Technologies, Waltham, MA, USA), 0.4 µL 5′ primer (1 µM), 0.4 µL 3′ primer (2 µM), 2 µL purified PCR products, and 6μL ddH2O which was combined in these conditions: 38 cycles at 94 °C for 1 min, then at 56 °C which diluted the combined reaction product (0.5 µL), followed by mixing with 0.5 µL L Liz500 size standard, 9 µL Hi-Di (Applied Biosystems, Foster City, CA, USA), and denaturing at 95 °C for 5 min. The sequencing process was performed using the ABI3730XL automatic sequencer (Applied Biosystems, Foster City, CA, USA). GeneMapper 4.1 (Applied Biosystems, Foster City, CA, USA) was used to analyze the raw data from the sequencer.

### 2.6. Statistical Analysis

IBM SPSS 27.0 statistical analysis software was used to eliminate missing data. The K-S test was used to determine the normal distribution. We used an independent *t*-test to compare obesity-related indicators between the normal weight group and the overweight/obese group. The single nucleotide typing of *TLR4 rs1928295* was verified by Hardy–Weinberg equilibrium using Pearson’s χ^2^ test. Binary logistic regression was used to compare over-weight/obesity and central obesity between genotypes. Then, 95% confidence intervals (95% CI) and odds ratios (OR) were also calculated. Based on 14 food groups, K-M-O test and Bartlett’s spherical test were used to test the feasibility of the data, and principal component factor analysis was used to identify dietary patterns. According to the lithotriptic diagram and the standard retention diet pattern with characteristic root greater than 1, the factor scores of each diet pattern were conducted in order from smallest to largest by the quart method, and the parts with the least tendency (Q1) and the most tendency (Q4) were selected for each diet pattern. Variance analysis was used to compare obesity-related indicators between *TLR4 rs1928295* genotypes, energy, and nutrient intake and dietary pattern factor scores. Multiple logistic regression was used to compare the influence of the interaction between different genotypes and dietary patterns on obesity indicators. A *p* value less than 0.05 was considered significant, and a *p* value between 0.05 and 0.1 was considered marginally significant.

## 3. Results

A total of 798 participants were included in the study after excluding null and invalid samples, of which there were 400 (50%) boys and 398 (50%) girls. The age span ranged from 7 to 12 years, with an average age of 9.2 ± 0.9. It was found that the incidence rate of general overweight/obesity and central obesity was 21% (boys 26% and girls 16%) and 16% (boys 19% and girls 13%), respectively.

### 3.1. Association between TLR4 rs1928295 Polymorphism and the Incidence Rate of Overweight/Obesity

In our study, the frequency of the C allele was 34.8% and the frequency of T allele was 65.2%. There were three genotypes: CC (12.8%), CT (44%), and TT (43.2%). As expected, the Hardy–Weinberg equilibrium did not deviate (*p* = 0.671). 

Based on logistic regression analysis, overweight/obesity rates were significantly higher (30.6%) in males with the TT genotype compared with CC/CT genotypes (20.3%) after adjusting for potential confounders (age, energy take) (OR 1.716; 95% CI 1.086~2.711; *p* = 0.038). Additionally, female children with the TT genotype had a significantly higher proportion of overweight/obesity (24.0%) than those with the CC/CT genotypes (17.5%) after adjusting for potential confounders (age, energy take) (OR 1.698; 95% CI 1.064~2.687; *p* = 0.043).

### 3.2. Comparison of Obesity Related Indicators between the CC + CT and TT Genotype of TLR4 rs1928295 Polymorphism

The results showed that boys with the TT genotype had a slightly higher waist circumference than those with the CC + CT genotype (*p* = 0.057). Girls with the TT genotype had a significantly lower level of diastolic blood pressure than those with the CC + CT genotype (*p* = 0.014) and a significantly lower level of high-density lipoprotein than those with the CC + CT genotype (*p* = 0.006) (Table 1).

### 3.3. The Relationship between Indicators Related to Obesity, Macronutrients Intake, and TLR4 rs1928295 Polymorphism

The following interactions were observed via a multiple linear regression analysis:(1)Protein intake and cholesterol intake had significant interactions with *TLR4 rs1928295* polymorphism in male children (*p*-interaction = 0.02 and 0.015). In this way, protein intake and cholesterol intake in the highest quartile group significantly increased LDL levels compared with protein intake and cholesterol intake in the lowest quartile group (*p* = 0.008 and 0.011) for the TT genotype, but there was no statistical difference in the CC/CT genotypes (*p* = 0.577 and 0.751) (Figure 1a,b).(2)There were significant interactions between total energy intake and the *TLR4 rs1928295* polymorphism in female children (*p*-interaction = 0.018). In this way, TG levels were significantly increased in the TT genotype (*p* = 0.003) for total energy intake in the highest quartile group compared to total energy intake in the lowest quartile group, while there was no statistical difference in the CC/CT genotypes (*p* = 0.08) (Figure 1c).

### 3.4. The Relationship between Children’s Dietary Patterns and Obesity-Related Indices

#### 3.4.1. The Relationship between Male Children’s Dietary Patterns and Obesity-Related Indicators

Four main dietary patterns of boys were determined by principal component factor analysis: healthy balanced dietary patterns (HBDP); a diet based on nuts and desserts (NDDP); a diet based on fish and pork (AFDP); and wheat-based dietary patterns (HWDP) (Table 2). In the HBDP, vegetables, fruits, tubers, and poultry were all included in a balanced diet. It was found in the NDDP that high-energy foods including nuts, sweets, and pastries were consumed in high amounts. A high level of animal foods, such as pork and fish, was found in AFDP. HWDP was characterized by a high wheat diet.

Boys predisposed to the HBDP were significantly associated with lower rates of central obesity (*p* = 0.045). There was a significant difference in LDL levels between boys predisposed to the AFDP before (*p* = 0.014) and after adjusting for potential confounders (*p* = 0.016). There was a relationship between AFDP boys and higher FBG levels before and after adjustment (*p* = 0.016 and *p* = 0.047, respectively). After adjustment (*p* = 0.004), HWDP was associated with lower FBG levels than before (*p* = 0.010) (Table 3).

#### 3.4.2. The Relationship between Female Children’s Dietary Patterns and Obesity-Related Indicators

Girls’ dietary patterns were divided into four groups based on the dietary data: dietary patterns based on pork, nuts, and wheat (NPDP); dietary patterns based on desserts and poultry meat (DPDP); dietary patterns based on eggs, fruits, and fish (EFDP); and dietary patterns based on vegetables and fish (VFDP) (Table 4). Nuts and pork consumption were higher in NPDP. There was a higher consumption of energy-dense foods like sweets and pastries in DPDP. The intake of fruits, eggs and fish was higher in EFDP. VFDP was characterized by a balanced meat and vegetable diet with a high intake of fish and vegetables. 

Girls who tended to be EFDP were significantly associated with lower rates of central obesity (*p* = 0.027) (Table 5).

### 3.5. Association of TLR4 rs1928295 Polymorphisms with Energy Intake and Dietary Patterns

On closer examination, it was found:(1)*TLR4 rs1928295* polymorphism and AFDP dietary pattern of male students had significant interaction with SP (*p* = 0.044), among which, the male students carrying the TT genotype who were most inclined to AFDP dietary pattern had higher SP (*p* = 0.0333). There was no significant difference between the male patients with the CC/CT genotype (*p* = 0.8323) (Figure 2a).(2)*TLR4 rs1928295* polymorphism had a significant interaction with AFDP dietary pattern in male students (*p* = 0.017), among which, the male students with the TT genotype who were most inclined to AFDP dietary pattern had a higher LDL (*p* = 0.0541). There was no significant difference in the male patients with the CC/CT genotype (*p* = 0.1496) (Figure 2b).

No significant influence of *TLR4 rs1928295* polymorphism on obesity-related indexes was found in female children under the interaction of dietary patterns.

## 4. Discussion

Children have different BMI and WHtR boundaries depending on their gender and age for obesity screening, and these boundaries are not uniform in different countries due to gender, age, and regional differences. Our study used the latest published screening standard for overweight and obesity in Chinese children, which makes our findings comparable among similar China-wide studies. Obesity-related indicators, such as blood pressure and lipids profile, have a significant correlation with obesity, which has been confirmed in our previous study [24], so this study continues to adopt these indicators for comprehensive analysis. 

One of this study’s attractive findings was the link between a higher obesity incidence and the TT genotype of *TLR4 rs1928295*. Although this locus was found to be significantly associated with BMI in European descent adults before [18], it was shown in Chinese Han children for the first time. This relationship will be confirmed in the future between carriers of the TT genotype and other obesity-related indices by further and larger sample investigation. It was also found that the TT genotype shows a minor high waist circumference and low high-density lipoprotein level, two marked characteristics of obesity [25,26]. Although these two indicators were not significantly different between genotypes, interesting results were found in the subsequent interaction analysis with energy nutrients and dietary patterns. 

Obesity is heritable and the majority of inherited susceptibility is related to the cumulative effect of many common DNA variants [27]. The related gene polymorphism can have an impact on the proportion of energy nutrient intake. For example, the GG genotype carriers of the *FTO* gene have a higher intake of carbohydrates [28]. The CC + CG genotype of *Cry1 rs2287161* increases fat intake in overweight obese women [29].

In this study, high protein and cholesterol intakes were significantly associated with elevated LDL levels in male children with the *TLR4 rs1928295* TT genotype; high energy intake was significantly associated with elevated TG levels in female children carrying the TT genotype. Protein can be divided into animal protein and plant protein, as they had different effects on obesity-related indices [30,31]. Obesity and overweight may be related to animal protein intake [32]. However, plant protein intake, especially soy protein intake was associated with lower LDL levels [33,34]. Our study only calculated total protein intake and further classification of animal protein and plant protein will be needed in the future, since they affect the body differently. Studies have shown that LDL levels can be reduced in children by reducing dietary fat and cholesterol intake [35]. A MEDIS study revealed that the energy intake was positively associated with the occurrence of hypertriglyceridemia [36], so limiting energy intake can be instrumental in controlling TG levels. Thus, our findings in the TT genotype were similar to those of the above studies which suggests that the TT genotype of *TLR4 rs1928295* may be a risk factor associated with obesity and lipids profile by affecting protein, cholesterol, and energy intake. 

Due to the complex interactions between nutrients and other dietary components, analyzing dietary patterns associated with childhood obesity is considered a more practical approach [37]. In this study, we identified four dietary patterns for male children and female children, respectively, to further analyze the association of different dietary patterns with obesity and related indicators, as well as their interaction with *TLR4 rs1928295* polymorphisms. 

For the dietary patterns of male children: (1) the HBDP significantly reduced the rate of central obesity and slightly reduced the rate of overweight/obesity. This dietary pattern is similar to the Mediterranean diet pattern which has been found to be associated with reduced obesity in both adults [38] and adolescents [39]. It is characterized by a predominance of vegetables and fruits, supplemented by animal foods. Vegetables are characterized by a high fiber content [40], which can effectively prevent the occurrence of obesity [41]. Previous studies have shown that fiber can interact with *FTO* gene polymorphism to reduce the risk of obesity [42]. A recent study found that folic acid, which is abundant in vegetables, can reduce the levels of several inflammatory cytokines in overweight and obese women who have the *MTHFR C677T* TT genotypes [43]. At the same time, vegetables are also one of the main sources of polyphenols [44], which play a role in regulating inflammatory cell signaling pathways [45]. It is further found that polyphenols can inhibit the overexpression of inflammatory mediators through the *TLR4* pathway [46,47], and obesity is known to belong to chronic low-grade inflammation [11]. Therefore, the combination treatment of polyphenols and *TLR4* polymorphism is likely to become one of the new ways to prevent and treat childhood obesity in the future. In addition, the interaction between polyphenol-rich apple juice and the CC genotype of *IL-6* (interleukin-6) *rs1800795* significantly reduced body fat content [48]. (2) The AFDP can significantly increase the levels of LDL and FBG. The AFDP is dominated by animal foods including fish and pork intake, which may need to be discussed in two separate ways. Fish are rich in n-3 PUFA (polyunsaturated fatty acids) [49], which has anti-inflammatory and antihypertensive effects [50]. Studies have shown that n-3 PUFA might play an anti-inflammatory role by attenuating the activation of the *TLR4* signaling pathway through saturated fatty acids, thereby reducing the risk of obesity [51]. However, on the other hand, the long-term consumption of pork can increase the burden on the pancreas [52] and lead to an increase in fasting blood glucose [53]. Recent studies have found that saturated fatty acids are associated with the increase in inflammation [54]. At the same time, it can activate the *TLR4* receptor [55], which further leads to the occurrence of obesity. (3) Furthermore, we found that the AFDP could interact with male, child carriers of the TT genotype of *TLR4 rs1928295*, resulting in increased LDL and SP levels. Studies have shown that saturated fatty acids can interact primarily with obesity-related genes, leading to weight gain compared to polyunsaturated fatty acids [56]. Saturated fatty acids in pork can significantly increase LDL levels [57] and blood pressure [58], and their content in pork is affected by the method of cooking [59]. So, pork rich in saturated fatty acids may play a major role in this AFDP.

For the dietary patterns of female children: (1) the NPDP significantly increased the levels of TG in our study. The NPDP is mainly composed of pork and nuts which may also need to be discussed in two separate ways. It has been found that the intake of red meat (including pork) is positively related to TG [60], and contributes to the increase in central obesity rate and overweight/obesity rate [61], while the intake of nuts is related to the decrease in TG and TC [62,63]. So, as with the AFDP in male children, pork is rich in saturated fatty acids and plays a major role in this NPDP. (2) The EFDP significantly reduced the rates of central obesity in our study. The EFDP focuses mainly on fruit, eggs, and fish. The intake of fruits is associated with the improvement of obesity indicators [64] and can reduce the risk of central obesity [65], while the intake of eggs can reduce hunger [66] and significantly reduce short-term energy intake in children [67]. Additionally, the advantages of fish have been fully stated above. So, this is a relatively healthy dietary pattern for obesity prevention. 

In conclusion, the TT genotype of *TLR4 rs1928295* is a potential risk factor for overweight/obesity and lipids profile in Chinese children. This SNP may be associated with related energy nutrient intake and eating patterns to some extent, but the relevant molecular mechanism was not involved in this study. The study on the association between TLR4 gene polymorphism and obesity is very rare at present, and this paper opens up a new territory for it. *TLR4* is the core of the mammalian innate immune system [68]. As a cell transmembrane receptor in the innate immune system, its mediated inflammatory signaling pathway is considered to be one of the main triggers of obesity-induced inflammatory response [69], and *TLR4* activity plays a prominent role in the occurrence of obesity-related inflammation [70]. Studies have shown that expression of the *TLR4* gene in obese patients is significantly higher than that in normal weight patients [71], which may be related to the fact that immune cells in obese individuals have a specific epigenetic profile [72], which may also be related to the increased number of TLR4 ligands in obese patients [16]. *TLR4* deficiency prevents diet-induced obesity [73], which has been found to be related to the reduced ability of fatty acids to induce inflammatory signaling in fat cells or tissues and macrophages [74]. As a natural ligand of *TLR4*, saturated fatty acids can be released in large quantities through macrophage-induced adipocyte lipolysis [75], and their mediated inflammatory effects need to be induced by the unknown endogenous ligand of *TLR4* [76]. This may explain why the interaction between the AFDP and *TLR4 rs1928295* in male children in this study led to the increase in obesity indicators. It can be seen that the *TLR4* gene builds a bridge between diet and obesity, and whether other *TLR4* gene polymorphisms have similar effects still needs more relevant studies in the future. As we know, the genetic factors of obesity are influenced by the combination of multiple genetic loci, and only by studying the genetic network of multi-gene–environment interaction from a systematic perspective is it possible to gradually elucidate the real causes of obesity occurrence. Nonetheless, the TT genotype of *TLR4 rs1928295* locus identified in this study has the potential to serve as a risk factor for predicting overweight/obesity in Chinese Han children aged 7–12 years, providing a new valuable clue for preventive interventions against childhood obesity.

Some limitations in our study should be noted. While BMI may be sensitive to changes in adiposity, it is a weak predictor of these changes in total body fat (%FAT) due to the poor specificity [77]. Moreover, it can lead to errors due to different ages and ethnic groups [78]. The latest study has showed that visceral adiposity index (VAI) can be used as a general predictor to identify metabolic disorders in Chinese children and adolescents [79]. Secondly, early onset of puberty is an influencing factor on BMI and blood lipids, especially for girls [80,81]. However, we did not carry out the Tanner staging measurements due to the privacy protection of children by schools and parents. Thirdly, causation cannot be established through a cross-sectional study, so we need to conduct further cellular and molecular biological experiments to find out the exact mechanism. In addition, it is not enough to attribute obesity risk solely to one SNP, the genetic influence on obesity is a complex factor, and GWAS demonstrated that it occurs through interactions with multiple SNPs with small effects and environmental factors. Our group has also been working to identify novel genetic loci associated with obesity. In the future, we will try to combine other different SNPs with external environmental factors for further research.

## 5. Conclusions

This study firstly demonstrates the association between a novel SNP, *TLR4 rs1928295*, and overweight/obesity in Han Chinese children aged 7–12 years. The TT genotype was probably a potential risk factor for obesity and the lipids profile. Subsequently, it was shown that the TT genotype can interact with increased intake of protein and cholesterol, which is related to higher LDL levels in boys, and can also interact with an increase intake of total energy, which is related to an increase in TG in girls. For dietary pattern: boys with HBDP predisposition had a lower incidence of overweight/obesity, while those with AFDP predisposition were associated with a high LDL level; girls with EFDP predisposition had a lower incidence of central obesity, but those with NPDP predisposition were associated with a higher TG level. Furthermore, the AFDP predisposed boys with the TT genotype were more likely than those with the CC + CT genotype to have an increase in LDL and SP levels. We hope that our study will provide new clues to improve the genetic etiological network of obesity and its interaction with dietary factors.

## Figures and Tables

**Figure 1 nutrients-15-03441-f001:**
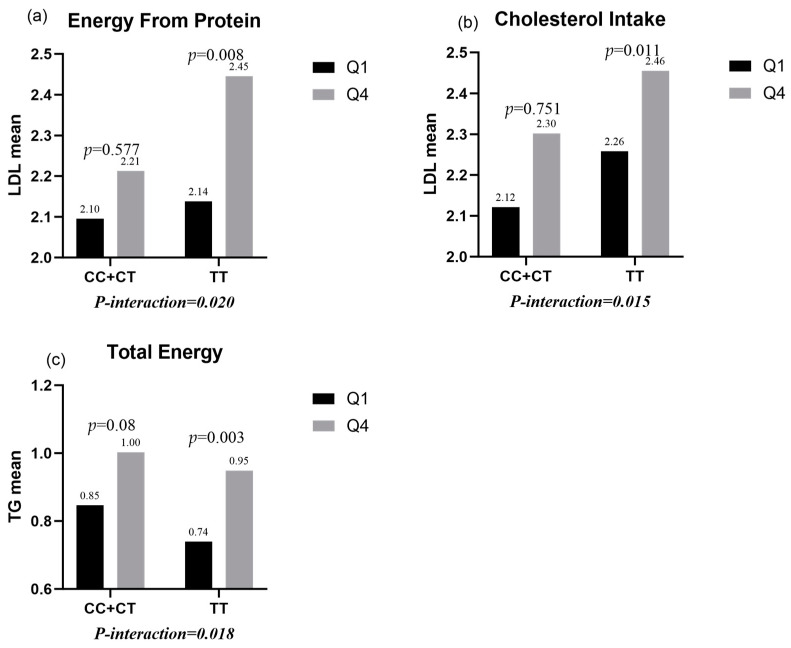
(**a**) The interaction between *TLR4 rs1928295* polymorphism in boys and protein intake on LDL; (**b**) the interaction between *TLR4 rs1928295* polymorphism in boys and cholesterol intake on LDL; (**c**) the interaction between *TLR4 rs1928295* polymorphism in girls and total energy intake on TG.

**Figure 2 nutrients-15-03441-f002:**
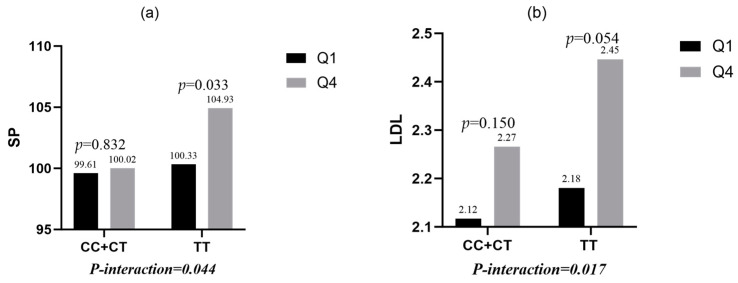
(**a**) The effect of the interaction between *TLR4 rs1928295* polymorphism and AFDP (animal foods including fish and pork-based dietary pattern) on SP. (**b**) The effects of the interaction between *TLR4 rs1928295* polymorphisms and AFDP on LDL.

**Table 1 nutrients-15-03441-t001:** The relationship between genotypes and obesity-related indicators.

Variables	Male			Female		
	CC + CT (*n* = 230)	TT (*n* = 170)	*p*-Value ^1^	CC + CT (*n* = 223)	TT (*n* = 175)	*p*-Value ^1^
BMI	17.33 ± 2.97	17.60 ± 2.96	0.608	16.42 ± 2.50	16.63 ± 2.64	0.272
WC	58.26 ± 7.85	59.28 ± 8.65	0.057	55.14 ± 6.71	55.60 ± 7.42	0.09
WHtR	0.43 ± 0.05	0.43 ± 0.05	0.472	0.41 ± 0.04	0.41 ± 0.05	0.306
SP (mmHg)	99.50 ± 9.72	101.12 ± 9.64	0.562	99.26 ± 9.89	99.16 ± 9.33	0.249
DP (mmHg)	61.83 ± 6.19	62.12 ± 6.54	0.423	61.76 ± 6.88	61.55 ± 5.64	0.014
TC (mM)	3.84 ± 0.76	3.82 ± 0.81	0.546	3.93 ± 0.85	3.80 ± 0.84	0.56
TG (mM)	0.84 ± 0.53	0.81 ± 0.63	0.62	0.86 ± 0.49	0.84 ± 0.42	0.265
HDL (mM)	1.46 ± 0.29	1.41 ± 0.26	0.147	1.41 ± 0.24	1.39 ± 0.28	0.006
LDL (mM)	2.24 ± 0.58	2.27 ± 0.64	0.903	2.33 ± 0.68	2.27 ± 0.62	0.56
FBG (mM)	4.73 ± 0.44	4.75 ± 0.44	0.799	4.64 ± 0.40	4.64 ± 0.36	0.321

Note: ^1^ Independent *t*-test. Abbreviations: BMI (body mass index), WC (waist circumference), WHtR (waist-to-height ratio), SP (systolic pressure), DP (diastolic pressure), TG (triglyceride), TC (total cholesterol), HDL (high density lipoprotein), LDL (low density lipoprotein), FBG (fasting blood glucose).

**Table 2 nutrients-15-03441-t002:** Factor loads of four identified dietary patterns in boys.

Food Species	Dietary Patterns			
	HBDP	NDDP	AFDP	HWDP
Vegetables	0.758			
Fruits	0.580			
Fish			0.708	
Egg	0.476			
Dairy				
Rice				−0.723
Wheat				0.726
Potato	0.484			
Nut		0.769		
Pork			0.708	
Poultry meat	0.402			
Cakes and pastries		0.684		
Candies		0.444		
Beans and their products				
Characteristic root	1.820	1.430	1.393	1.382
Variance (%)	12.997	10.216	9.953	9.630
Variance of involvement (%) = 43.039				

**Table 3 nutrients-15-03441-t003:** Obesity Related Characteristics by quartile (Q) of dietary patterns for boys.

Variables	HBDP		*P* ^3^	*P*′ ^4^	NDDP		*P* ^3^	*P*′ ^4^	AFDP		*P* ^3^	*P*′ ^4^	HWDP		*P* ^3^	*P*′ ^4^
	Q1 (*n* = 100)	Q4 (*n* = 100)			Q1 (*n* = 100)	Q4 (*n* = 100)			Q1 (*n* = 100)	Q4 (*n* = 100)			Q1 (*n* = 100)	Q4 (*n* = 100)		
Age (year)	9.41 ± 0.94	9.35 ± 1.00	0.699		9.58 ± 0.84	9.131 ± 0.87	<0.001		9.20 ± 0.95	9.50 ± 0.89	0.023		9.24 ± 0.93	9.41 ± 0.88	0.193	
Overweight/obesity (%) ^1^	16.50%	10.50%	0.056		15.50%	13.50%	0.533		13.50%	16.50%	0.355		13.00%	13.50%	0.873	
Central obesity (%) ^1^	12.00%	6.50%	0.045		10.00%	7.00%	0.259		8.50%	10.00%	0.585		9.50%	8.50%	0.713	
SP (mmHg) ^2^	100.36 ± 9.09	100.32 ± 10.01	0.976	0.952	100.45 ± 9.52	99.76 ± 10.07	0.619	0.614	99.89 ± 10.08	102.18 ± 10.41	0.116	0.247	100.98 ± 9.78	99.72 ± 8.88	0.341	0.205
DP (mmHg) ^2^	61.80 ± 6.34	61.46 ± 5.84	0.694	0.730	62.34 ± 5.85	62.40 ± 6.35	0.945	0.422	62.58 ± 0.54	62.11 ± 6.25	0.604	0.483	62.19 ± 6.35	61.97 ± 6.51	0.809	0.713
TCH (mM) ^2^	3.89 ± 0.67	3.76 ± 0.95	0.281	0.277	3.81 ± 0.79	3.77 ± 0.83	0.757	0.999	3.79 ± 0.68	3.82 ± 0.80	0.821	0.932	3.90 ± 0.63	3.72 ± 0.81	0.080	0.062
TG (mM) ^2^	0.76 ± 0.43	1.02 ± 0.80	0.005	0.003	0.84 ± 0.57	0.92 ± 0.73	0.439	0.246	0.80 ± 0.46	0.98 ± 0.84	0.073	0.136	0.79 ± 0.46	0.83 ± 0.53	0.510	0.566
HDL (mM) ^2^	1.42 ± 0.25	1.43 ± 0.32	0.780	0.786	1.42 ± 0.26	1.46 ± 0.26	0.302	0.260	1.44 ± 0.27	1.45 ± 0.27	0.800	0.720	1.45 ± 0.27	1.44 ± 0.28	0.866	0.873
LDL (mM) ^2^	2.26 ± 0.54	2.28 ± 0.66	0.810	0.815	2.24 ± 0.68	2.24 ± 0.59	0.973	0.778	2.14 ± 0.50	2.35 ± 0.65	0.014	0.016	2.25 ± 0.60	2.20 ± 0.54	0.515	0.393
FBG (mM) ^2^	4.83 ± 0.37	4.75 ± 0.46	0.184	0.201	4.76 ± 0.44	4.71 ± 0.45	0.419	0.952	4.65 ± 0.42	4.80 ± 0.44	0.016	0.047	4.81 ± 0.43	4.65 ± 0.44	0.010	0.004

Note: ^1^ Categorical variables are expressed as percentages and ^2^ continuous variables are expressed as mean ± standard deviation (SD). The *p* values of ^3^ continuous variables (ANOVA) and category variables (Chi-square test) were ^4^ adjusted for age and total energy intake in ANCOVA model. Q1 (lowest) and Q4 (highest) represent the quartile of dietary patterns. Q1 represents the segment least inclined to a particular dietary pattern and Q4 represents the segment most inclined to a particular dietary pattern. Abbreviation: HBDP (Healthy Balanced Diet), NDDP (Diet based on Nuts and Desserts), AFDP (Diet based on fish and pork), HWDP (Diet based on wheat).

**Table 4 nutrients-15-03441-t004:** Factor loads of four definite dietary patterns in female children.

Food Species	Dietary Patterns			
	NPDP	DPDP	EFDP	VFDP
Vegetables				0.678
Fruits			0.665	
Fish			0.593	0.748
Egg			0.703	
Dairy				
Rice				
Wheat	0.573			
Potato				
Nut	0.718			0.388
Pork	0.757			
Poultry meat		0.529		0.387
Cakes and pastries		0.785		
Candies		0.649		
Beans and their products				
Characteristic root	1.618	1.580	1.462	1.420
Variance (%)	11.555	11.285	10.441	10.145
Variance of involvement (%) = 43.425				

**Table 5 nutrients-15-03441-t005:** Obesity Related Characteristics by quartile (Q) of dietary patterns for girls.

Variables	NPDP		*P* ^3^	*P*′ ^4^	DPDP		*P* ^3^	*P*′ ^4^	EFDP		*P* ^3^	*P*′ ^4^	VFDP		*P* ^3^	*P*′ ^4^
	Q1 (*n* = 100)	Q4 (*n* = 99)			Q1 (*n* = 100)	Q4 (*n* = 100)			Q1 (*n* = 100)	Q4 (*n* = 99)			Q1 (*n* = 100)	Q4 (*n* = 99)		
Age (year)	9.09 ± 0.95	9.35 ± 0.83	0.044		9.09 ± 0.95	9.25 ± 0.93	0.228		9.15 ± 0.90	9.06 ± 0.91	0.498		9.16 ± 0.80	9.27 ± 0.91	0.337	
Overweight/obesity (%) ^1^	9.50%	10.10%	0.831		7.50%	9.00%	0.568		13.00%	8.00%	0.083		8.50%	9.50%	0.688	
Central obesity (%) ^1^	10.10%	7.00%	0.272		6.50%	8.00%	0.547		10.00%	4.50%	0.027		3.50%	8.50%	0.028	
SP (mmHg) ^2^	97.42 ± 8.63	99.69 ± 10.22	0.092	0.244	98.54 ± 9.40	101.10 ± 10.30	0.068	0.102	100.79 ± 10.69	98.75 ± 10.00	0.165	0.191	99.65 ± 9.75	98.41 ± 8.84	0.350	0.268
DP (mmHg) ^2^	60.88 ± 5.92	62.12 ± 6.84	0.173	0.386	61.43 ± 6.54	62.54 ± 6.21	0.220	0.281	62.68 ± 6.30	61.57 ± 6.25	0.212	0.237	62.00 ± 6.37	61.70 ± 6.35	0.737	0.619
TCH (mM) ^2^	3.84 ± 0.72	3.72 ± 0.86	0.286	0.336	3.87 ± 0.86	3.92 ± 0.89	0.693	0.613	3.91 ± 0.93	3.86 ± 0.93	0.657	0.643	3.91 ± 0.95	3.87 ± 0.79	0.781	0.817
TG (mM) ^2^	0.82 ± 0.41	0.95 ± 0.50	0.036	0.053	0.89 ± 0.53	0.89 ± 0.45	0.984	0.924	0.85 ± 0.45	0.91 ± 0.48	0.343	0.36	0.82 ± 0.42	0.82 ± 0.46	0.952	0.958
HDL (mM) ^2^	1.37 ± 0.28	1.41 ± 0.25	0.311	0.309	1.39 ± 0.27	1.40 ± 0.27	0.887	0.862	1.39 ± 0.26	1.43 ± 0.27	0.327	0.307	1.44 ± 0.26	1.37 ± 0.26	0.048	0.055
LDL (mM) ^2^	2.29 ± 0.58	2.27 ± 0.62	0.888	0.981	2.25 ± 0.66	2.39 ± 0.72	0.157	0.115	2.31 ± 0.67	2.38 ± 0.74	0.527	0.577	2.41 ± 0.72	2.27 ± 0.62	0.152	0.161
FBG (mM) ^2^	4.66 ± 0.40	4.64 ± 0.38	0.752	0.542	4.63 ± 0.35	4.71 ± 0.37	0.148	0.223	4.70 ± 0.43	4.60 ± 0.40	0.094	0.116	4.56 ± 0.37	4.65 ± 0.40	0.097	0.146

Note: ^1^ Continuous variables are expressed as mean ± standard deviation (SD) when ^2^ categorical variables are expressed as percentages. The *p* values of ^3^ continuous variables (ANOVA) and category variables (Chi-square test) were ^4^ adjusted for age and total energy intake in ANCOVA model. Q1 (lowest) and Q4 (highest) represent the quartile of dietary pattern. Q1 represents the segment least inclined to a particular dietary pattern and Q4 represents the segment most inclined to a particular dietary pattern. Abbreviations: NPDP (pork, nut, and wheat-based diet), DPDP (pastry and confectionery based diet), EFDP (fruit, egg, and fish-based diet), VFDP (vegetable and fish-based diet).

## Data Availability

Due to the nature of this research, participants of this study did not agree for their data to be shared publicly, so supporting data is not available.

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
