# Peer review of "TT Genotype of TLR4 rs1928295 Is a Risk Factor of Overweight/Obesity in Han Chinese Children Aged 7–12 Years and Can Interact with Dietary Patterns to Affect the Incidence of Central Obesity and Lipid Profile, Systolic Blood Pressure Levels"

_nutrients, 2023, doi:10.3390/nu15153441_

Round 1

Reviewer 1 Report

The aim of the authors was to evaluate the interaction between Dietary patterns, TLR polyphormism, obesity and factors related to obesity. The study is interesting, but it has some issues that must be improved and clarified

Line 20-21 - Please review this result. The description does not seem to be correct.

Line 21- Rats?

Please improve the abstract. In conclusion the authors report: "Which is Somewhat Correlate with Dietary Patterns." However, the authors described very little about the Dietary Patterns in the results to support this conclusion.

- Please review the use of the word "interaction", because, at times, it was used in a mistaken way.

Line 57-66 - Improve this paragraph. What is TLR4? How does it relate obesity? And what is the importance of using this marker and not another?

Line 84- Replace Reaearch with Research

Line 112-114 - What protocol was used to measure blood pressure?

Line 120 - Why did you do the records on consecutive days?

Line 127 - Explain better how eating patterns were defined.

- Why did you use two different statistics software?

Line 194-195 - If they are different why is the percentage the same? Please correct.

Table 1 - Does P refers to logistics regression or square qui test? Also, the or represents the risk of obesity to which genotype? Please better detail how this analysis was done in the caption and describe the result of the separate odds ratio from the chi-square result.

The table is not a correlation table as it indicates in the title.

Line 204 - A P = 0.09 does not reproach a tendency for the authors to use the term "A Slightly Higher"

Line 215- In this topic there is no interaction assessment. Please remove this term from the subtitle

Figure 1 - What statistical analysis was performed?

Why did the authors present two different ways to analyze the same parameters (Figure 1 and Table 4)? You should only choose one of them.

The title does not match what author shows, because the interaction was seen between Dietary Patterns, Polymorphism and Indicators related to obesity and not the risk of obesity

- The results do not show what is described in the title: the interaction between the three factors: diet, obesity and genetics.

Discussion: Please improve the discussion of results. This section is another repetition of the results and little discussion is made.

Reviewer 2 Report

The aim of this study is to test the relationships between the SNP rs1928295 of the TLR4 gene and diet, obesity, and parameters of health status in Han Chinese children. The authors found some interesting results, but several revisions are needed. In general, it is not clear why the authors study this gene. This must be clarified in the Introduction and in the Discussion sections. In addition, there are numerous oversights in the text.

Abstract:

-A brief section about the background is needed. Particularly is important to describe why the authors start the study of this gene and polymorphism.

Introduction:

-Lines 37-39: This sentence is not clear. Race, gender, living environment, and eating habits are not genetic factors.

-Lines 57-61: This sentence is too long and not clear. In particular, is important to elucidate the role of TLR4 gene. The authors have to include the entire name of the gene and specify the role of the SNP (intronic? Missense?...)

Materials and Methods:

-Line 24: What is SY? Please explain it.

-“Anthropometric Assessments” Paragraph: How were identified the overweight/obese subjects? It is not clear. 

-Line 126: What is CDC?

-Line 171: two dots at the end of the sentence. Please correct.

Results: 

-A brief description of the sample is necessary (number of participants, percentage of males/females, mean age, mean BMI, etc). All the variables included in the Results section must be described. A table with the sample characteristics could be useful. 

-Lines 188/189: The sentence is not necessary since the allelic frequencies are reported in lines 186/187.

-Lines 190 and 195: I think ~ is not correct for the confidence interval.

-It is not clear why the authors in order to verify the association between continuous variables (BMI etc) did not use linear regression but T-test. 

-Table 6 is difficult to read.

Round 2

Reviewer 2 Report

I thank the authors for the changes. I have just a little suggestion.

- The sentence "Race, gender and even individuals have different susceptibility to these factors[5], which provide new insights into 41 the susceptibility to obesity. " is not clear and not useful. I suggest removing it.

Author Response

The sentence "Race, gender and even individuals have different susceptibility to these factors[5], which provide new insights into 41 the susceptibility to obesity. " has been removed. 

Thanks for your suggestion.
